# Investigation of a Deep Brain Stimulator (DBS) System

**DOI:** 10.3390/bioengineering10101160

**Published:** 2023-10-03

**Authors:** Jennifer Whitestone, Anmar Salih, Tarun Goswami

**Affiliations:** 1Department of Biomedical, Industrial and Human Factors Engineering, Wright State University, Dayton, OH 45435, USAsalih.7@wright.edu (A.S.); 2Department of Orthopedic Surgery, Sports Medicine and Rehabilitation, Miami Valley Hospital, Dayton, OH 45409, USA

**Keywords:** leads, deep brain stimulation, extensions, impedance

## Abstract

A deep brain stimulator (DBS) device is a surgically implanted system that delivers electrical impulses to specific targets in the brain to treat abnormal movement disorders. A DBS is like a cardiac pacemaker, but instead of sending electrical signals to the heart, it sends them to the brain instead. When DBS leads and extension wires are exposed in the biological environment, this can adversely affect impedance and battery life, resulting in poor clinical outcomes. A posthumously extracted DBS device was evaluated using visual inspection and optical microscopy as well as electrical and mechanical tests to quantify the damage leading to its impairment. The implantable pulse generator (IPG) leads, a component of the DBS, contained cracks, delamination, exfoliations, and breakage. Some aspects of in vivo damage were observed in localized areas discussed in this paper. The duration of the time in months that the DBS was in vivo was estimated based on multiple regression analyses of mechanical property testing from prior research of pacemaker extensions. The test results of three DBS extensions, when applied to the regressions, were used to estimate the in vivo duration in months. This estimation approach may provide insight into how long the leads can function effectively before experiencing mechanical failure. Measurements of the extension coils demonstrated distortion and stretching, demonstrating the changes that may occur in vivo. These changes can alter the impedance and potentially reduce the effectiveness of the clinical treatment provided by the DBS system. Ultimately, as both DBSs and pacemakers use the same insulation and lead materials, the focus of this paper is to develop a proof of concept demonstrating that the mechanical properties measured from pacemaker extensions and leads extracted posthumously of known duration, measured in months while in vivo, can be used to predict the duration of DBS leads of unknown lifespan. The goal is to explore the validity of the proposed model using multiple regression of mechanical properties.

## 1. Introduction

Deep brain stimulation (DBS) represents a life-changing modality for patients with movement disorders. The basal ganglia, a group of nuclei in the brain, are responsible for the body’s motor control. However, when the body is unable to regulate the chemical and electrical functions of the brain’s neurons, movement disorder symptoms may result in essential tremors, dystonia, bradykinesia, and rigidity [1,2,3,4]. A treatment option delivered via a DBS system is used to reduce the symptoms of ailments such as Parkinson’s disease, epilepsy, obsessive compulsive disorder (OCD), dystonia, and essential tremors, as well as others. Figure 1 is a pie chart showing the percentage of the population with these disorders that are utilizing DBS as a treatment for movement symptoms [5].

Figure 1 shows that 61.12% of patients with Parkinson’s disease are treated with DBS devices, making this the predominant population utilizing this therapeutic modality. Additionally, 22.64% of the population afflicted with essential tremors utilizes DBS to control the movement disorder associated with this disease. Other movement disorders treated using DBS make up the remaining 16.24%.

The DBS intrinsic features include the integrated pulse generator (IPG) that acts as the waveform generator and power source, DBS leads (electrodes) that are implanted in the brain tissue, and the extensions that connect the IPG to the leads [1]. Often referred to as the pacemaker for the brain, the DBS system delivers electrical impulses to reduce abnormal movements through electrodes surgically implanted into the brain tissue that are powered by an IPG programmable element. An image of the DBS system components can be seen in Figure 2.

The IPG contains a battery, a power module, a computer processing unit (CPU), and a microprocessor that manages all of the device’s functions, including activation, deactivation, pulsing parameters, internal diagnostics, and communication with external devices [6]. The IPG sends electrical signals via extensions to the electrode leads that, when operating correctly, reduce disease symptoms. The electrode lead is a thin insulated wire, which is surgically inserted through a small opening in the skull. The lead is implanted within a targeted brain area based on MRI to address and minimize movement symptoms. This is accomplished by delivering current to nearby nerve fibers and cell bodies with an electrical field. Since the neurostimulator controls the flow of current to specific brain regions, treatment parameter configurations can be optimized by considering pulse widths, frequencies, amplitudes, battery life, and potential lead migration. The extension is an insulated wire passed under the skin of the head, neck, and shoulder, connecting the lead to the IPG.

If the DBS system unexpectedly ceases to function due to an incident such as an electrical short or open circuit, conductor wire fracture, or insulation breach, the mobility symptoms can return. Figure 3 is a chart that shows the component performance failures that have been found to occur in DBS systems. As with most medical devices, even small deviations in any part of the system can compromise treatment outcomes. It is the extension component that is of particular interest as these insulated wires must carry the signals required to safely deliver proper treatment to the brain and maintain the longevity of the DBS system. However, limited data are available on the degradation of insulated leads and extensions as these materials perform over time. Figure 3 indicates that extensions are the cause of failure in 19.2% of cases and leads are the cause in 46.1%, revealing that these insulated components contribute to the majority of failures of DBS systems.

As both DBS systems and pacemakers use the same insulation and lead materials, the focus of this paper is to develop a proof of concept demonstrating that the mechanical properties measured from pacemaker extensions and leads extracted posthumously of known duration measured in months while in vivo can be used to predict the duration of DBS leads of unknown lifespan. The goal is to explore the validity of the proposed model using multiple regression of mechanical properties.

## 2. Materials

Wright State University (WSU)’s Boonshoft School of Medicine Anatomical Gift Program provided the Biomedical Engineering Department of WSU both pacemakers and DBS systems which were extracted posthumously. The focus of this research is to identify evidence of possible failure modes such as erosion, fractures, cracks, abrasion, delamination, lead fractures, case damage, coil damage, and insulation attrition. Furthermore, this study proposes that mechanical properties measured from pacemaker leads extracted posthumously of known duration in vivo can be used to predict the duration of DBS leads of unknown lifespan.

A Medtronic DBS Soletra model 7426 extracted posthumously was examined to identify aspects leading to device failure. In evaluating the hardware-related complications of DBS, it is useful to consider the electrode leads, the extension wires, and the IPG as a system. The dysfunction of any one component can lead to system failure.

The IPG case is made of titanium alloy and houses the battery, power module, microprocessor, and additional communication electronics. The case is 60 mm in length with a 10 mm thickness.

The stretch-coil extensions connect the IPG and leads and are coated with silicone [7]. The insulation material, silicone, is used in medical applications due to its biocompatibility and electrical insulation properties. It provides a protective layer around the wires, preventing electrical interference and ensuring the safe and reliable transmission of electrical impulses from the electrodes to the pulse generator. It is placed under the skin and runs from the scalp, behind the ear, down the neck, and to the chest. The design of the extension with silicone coating is meant to be flexible and durable, allowing the extensions to be implanted under the skin and withstand the movements and stresses of everyday activities without compromising the integrity of the system. A number of observations and measurements can be conducted to interrogate the stability of the leads in their current state. Physical defects measured from the extracted components may be indicative of changes in the impedance of the system that occurred while in vivo. It is critical to identify potential sources of unstable impedance as this may lead to a lack of appropriate DBS or pacemaker therapy [8].

The lead is a coated wire with a number of electrodes at the tip that deliver electrical pulses to the brain tissue. It is placed inside the brain and connects to an extension wire through a small hole in the skull and is not examined in this study [9,10].

## 3. Methods

As the potential malfunction of leads and extensions constitutes the majority of performance events leading to device failure, the components’ lifespan is critical for maintaining optimal system performance.

Previous investigations of retrieved devices have resulted in damage assessments of DBS systems, documenting methods for measuring components [11,12]. Prior research conducted at WSU contributed to an understanding of extension and lead mechanical characteristics used to develop a predictive model of lead lifespan [11]. The researchers investigated the residual properties of silicone (MED-4719) leads used in cardiac implantable electronic devices, including pacemakers, defibrillators, and neuro-stimulators. The objective was to compare the properties of leads with varying in vivo exposure times and to evaluate their mechanical behavior. Leads of known duration extracted from patients and new leads were tested for mechanical properties, including load to failure, percentage elongation, ultimate tensile strength, percentage elongation at 5 N, and modulus of elasticity. The study findings indicated that the load to failure, elongation to failure, ultimate tensile strength, and percentage elongation at 5 N exhibited a significant decrease after specific durations of in vivo exposure. The extensions exhibited a higher modulus of elasticity over time in vivo, particularly after 71 months [11]. The data reported from this previous study were used to develop a predictive model for the leads of unknown exposure in the present study.

Additionally, the IPG and extension wires were investigated to quantify defects or damage using an Elikliv optical microscope and a Mitutoyo micrometer, while axial stiffness was recorded using a Test Resources Q series system, an electromechanical test device. An initial look at the IPG case, connector block, and extension wires revealed scratches, abrasions, pitting, separation of components, eruption and loss of insulation, and evidence of biomaterial deposits. Examples of images acquired using the digital microscope at up to 1000 times magnification are shown in Figure 4. Corrosion from the body’s environment results in pitting, holes, loss of insulation, and other physical damage including kinking, lead separation, abrasions, and scratches. This type of damage can potentially be found on all DBS components.

A Mitutoyo micrometer was used to measure the IPG case dimensions which were found to be well within 1 mm of the manufacturing specifications and showing no obvious signs of deformation. Figure 5 shows one of the five scratches seen using the Elikliv optical microscope. The lengths of the scratches were measured using the micrometer, with a result of an average of ~10 mm in length. The average width of the scratches was approximately 0.58 mm, consistent with the width of a medical scalpel most likely used during extraction.

The lithium–chloride battery life is estimated to be 4–5 years; however, this has been found to be shorter given variations in amplitude, pulse width, and daily use [13]. In fact, the battery is designed to provide continuous current down to 3.5 V, and the expected battery life averages 37.4 +/− 17.3 months, often less than desired. While battery life has been evaluated in previous studies, it is not within the scope of this investigation to measure the battery performance [14].

As shown in Figure 5, it is evident that the leads are separating from the connector block. Continuity tests using a Circuit Test Electronics multimeter were conducted and it was determined that no breakage existed from the leads at the point where the connector block was separating to the end of each of the coiled wires.

Digital microscope images of the leads revealed corrosion, cracks, pitting, eruptions, and tears (Figure 6). As these injurious damages to the insulation suggest that the extensions were stressed, an inverted compound microscope was used to measure the position and spacing of the underlying coils. Three sets of measurements (four each) were conducted using different methods to determine if the direction and position of the instrument’s light source influenced the ability to measure the coil. The dimensions were measured from left-to-left, middle-to-middle, and right-to-right as shown in Figure 7, maintaining the vector of measurement perpendicular to the coil, with an average difference of 7 µm. The coils did not appear to be stretched or warped in these locations. To determine if the coil thickness had been impacted by the damage to the insulation, the thicknesses were measured at two different locations and sampled five times each (Figure 8). The average thickness was 28 µm with a standard deviation of 4 µm, indicating that the coils did not undergo any distortion in these areas even though the insulation had been disrupted.

Lead and extension wire insulation can degrade over time, which is evidenced by discoloration and loss of physical properties given influences such as moisture, heat, and light. The length of time the DBS system and extensions had been used is unknown; therefore, an evaluation of the stiffness properties was performed and the duration of the lead life was estimated.

Using the Test Resources Q series system and referencing the ASTM standard D 412-06a, tensile testing was conducted to determine the mechanical strength of the insulated extensions. Previous studies demonstrated a reduction in axial strength and additional changes in mechanical properties in posthumously extracted leads [8,11,12]. For this investigation, the Medtronic Soletra system was extracted with 386 mm of lead, only some of which were impacted by erosion and other eruptions in the insulation. Three separate specimens were cut from the lead, using only those elements that showed no damage. The specimens were all cut to a length of 38 mm as shown in Figure 9. To prevent the lead from slipping out of the Test Resource clamps, small portions of 80-grit sandpaper were cut and folded over both ends of the leads. To allow a consistent 22 mm of test material between the Test Resource clamps, 8 mm at the end of each lead was secured using the 80-grit sandpaper. The tests were performed by applying specific loads on the samples and the insulation with coiled wires was stretched to failure. The test was repeated using two additional specimens cut to the same length and secured using the sandpaper, and an average was taken of the three measurements. All three tests demonstrated that the axial strength values were substantially less than those of a new system, indicating a level of degradation over time for this specimen. The load vs. displacement results from a sample are shown in Figure 10.

## 4. Results

Salih and Goswami [11] investigated 46 Medtronic leads of known duration (in months) by measuring five independent variables on each specimen including load to failure, maximum elongation, elongation at a load of 5 N, ultimate tensile strength (UTS), and modulus of elasticity. Linear regressions were developed given these variables to predict the duration of the system in months. For each of the variables, a significant difference was identified at different duration intervals. For instance, the UTS demonstrated a significant difference before and after 94 months. The R^2^ values, when accounting for differences in duration per variable, were quite good, ranging from 0.7 to 0.9. When the regressions are used without distinguishing between duration values, but across all 46 specimens, the R^2^ values range from 0.2 to 0.6 for the ability to predict duration as shown in Table 1.

The level of correlation using the measured mechanical properties independently suggests that in order to predict duration, a multivariate relationship among the variables is needed to develop a more accurate predictive algorithm. A principal components analysis (PCA) of the published WSU data, setting a threshold of explained variance of 95% and using all five factors, resulted in the first three components representing a total of 90.96% of the variability in the data. While the 1st component accounts for the most variation in the data, from small to large in the first four variables, the 2nd component shows an inverse relationship between load to failure and UTS vs. maximum elongation and elongation at 5 N. Table 2 shows the relationships among the five variables and the variance explained within the components.

A multiple regression analysis was conducted using the five independent variables (load to failure, maximum elongation, elongation at 5 N, UTS, and modulus of elasticity), and using duration as the dependent variable. The resulting equation to predict the duration is as follows:Duration = 232.1554 − (6.4647 ∗ Load to Failure) − (0.6782 ∗ Max Elongation) + (0.7357 ∗ Elongation at 5 N) − (0.2776 ∗ UTS) + (2.2391 ∗ Modulus of Elasticity)

When applying this equation to the Salih data [11], the coefficient of determination for this model is an R^2^ of 0.76 as shown in Figure 11. The plotted data show the relationship between measured and predicted duration. The linear regression is shown as the red line correlating the two variables. Not allowing for negative values when predicting the duration in months increases the R^2^ to 0.8. Therefore, for predicting the duration, the above model shows the relationship between the independent variables and the improved ability to derive the duration of the extension in vivo.

The characteristics of the extension samples tested in this study using the tensile tests resulted in values for load to failure, elongation to failure, percentage elongation at 5 N, UTS, and modulus of elasticity (Table 3). The values are compared to the range of values measured in the Salih study and fall within the ranges of the previous specimens. Using the linear regressions and the multiple regression equations independently, the duration in months was calculated for the three tested leads in this investigation for which the actual duration was unknown.

Using the Salih linear regression results of values ranging from 6.9 to 113.2 months provided a possible predicted duration of 106.3 months, as shown in Table 4. Applying the multiple regression equation, the predicted duration in months values ranged from 55.2 to 89.3 months as shown in Table 5, resulting in a span of possible predicted duration in months of 34.1. Using all five measured mechanical properties reduces the estimated range of the extension lifespan by 72.2 months.

By examining the coiled wires before and after the tensile tests, it is clear that stretching the extensions can potentially change the impedance of the system. Using the Elikliv electronic microscope, the impedance levels of the coiled wires were captured before and after stretching. As shown in Figure 12, the “stretched” coils are no longer arranged as circumferential elements but are elongated, separated, and create a narrower lead. The multimeter was used to measure four strands each of “stretched” and “original” coils exactly 32 mm in length from clip to clip. The average resistance of the original coiled wires was 2.3 ohms while that of the “stretched” wires averaged 1.4 ohms, indicating a lower resistance in leads that may have undergone stretching in vivo. The implications of stretched wires can be quite serious. If the leads are stretched due to repeated neck and head movements, the impedance could be reduced, the current flow increased, and the battery life decreased, potentially reducing the efficacy of the DBS system.

In fact, several models have demonstrated that impedance directly impacts the volume of tissue stimulated (VTS) [15,16]. Butson et al. modeled various impedance levels using a 3D model of the Medtronic 3387 DBS system to show that the volume of tissue activated increases volumetrically as the impedance decreases, indicating that the extension integrity and lead functions must be closely monitored [16].

## 5. Discussion

A number of observations were made from this evaluation of a posthumously extracted DBS system. It is possible to measure and record the external damage to the IPG and leads using an optical microscope and associated software to quantify corrosion, scratches, abrasions, tears, and pitting. It was proposed that the damage on the IPG cover was most likely post-extraction as the size of the scratches appeared to be the same as that of a scalpel or knife edge. While the connector block was separating from the IPG case, the continuity of the leads appeared to be intact. The load to failure, maximum elongation, percentage elongation at 5 N load, ultimate tensile strength, and modulus of elasticity were measured for the three samples and compared to previous measurements [11]. Using the data previously measured by Salih and Goswami, a multiple regression model was developed to predict the age of the extracted extension of the DBS system. The measured extension characteristics predicted a duration in months ranging from 55.2 to 89.3 months, narrowing the estimated duration range by 72.2 months with an R^2^ value of 0.8. Additional samples from the original lead could be measured to add confidence to this prediction. The resistance values of the post-tensile-test wire samples were measured and found to be considerably lower than those of the pre-stressed leads.

Evidence of lead breakage, while possibly a result of the component extraction, has been found to occur in vivo near the connection of the electrode with the extension wire. This is particularly true of those with the electrode lead wire running along the supraclavicular area, suggesting repeated head turning can fatigue and ultimately weaken or even break the extensions [17,18]. If the metal contacts within the extensions are exposed to body fluids, corrosion can occur. Additional studies have reported the prevalence of hardware failures including high impedance and fracture or failure of the lead or other parts of the implant [19]. Corrosion can also occur as a result of electrochemical reactions at the electrode–electrolyte interface [20]. Using scanning electron microscopy, researchers examined an explanted and damaged DBS electrode, capturing fissurations and cracks of the insulation tubing, degeneration of the internal core, and stretching of the wires. This prior study shows the qualitative and quantitative alterations of a malfunctioning lead and, to reduce the rate of hardware-related complications suggests the necessity of developing more reliable polymers [21]. Authors who retrospectively analyzed DBS systems of 43 ± 31 months determined hardware failure to be 2%, with the most important cause of lead fracture being the rotational movement of the lead–extension cable system [22]. When DBS leads are stretched, particularly to plastic deformation due to repeated neck and head movements, the impedance could be reduced, the current flow increased, and the battery life decreased, ultimately resulting in inappropriate treatment.

However miraculous the results of DBS can be, it is important to continue pursuing more robust materials and bio-resistive coatings. The complications of DBS systems include lead migration, infection, device failure, and device-related trauma. Failures include reimplantation of the IPG due to battery depletion, while other hardware failures include lead fraction, lead migration, extension wire, and IPG malfunction [23]. It is recommended that a staged approach to IPG replacement be considered, sparing the intact electrodes and extension wires, and supporting the idea of a holistic system evaluation when suspecting system failure [24]. Devices implanted long-term in the human body are subject to corrosion and, in the case of the leads, the impedance could possibly increase or decrease, changing the electrode properties. These changes, as well as battery failure, can impact the electrical charges that are ultimately injected into the neural tissue. Many potential complications and possible modes of failure exist with a DBS system that requires explantation. The usual practice is to remove and reimplant replacement leads after tissue healing, leaving patients without the clinical benefits of DBS for several months and, for some, at risk for DBS withdrawal, while some patients are no longer good surgical candidates for reimplantation. There exist methods to evaluate a DBS system in vivo such as radiofrequency ablation through the lead [25] and adaptive deep brain stimulation (aDBS) or closed loop neuromodulation systems, controlled by local field potentials (LFPs), that can prevent some repeat surgeries and postoperative risks [26,27]. Predictive models of the lifespan of implanted DBS leads and hardware can further guide clinicians in administering and optimizing patient treatment. New developments will continue to enhance and improve the delivery of electrical impulses to the brain including advancements in pattern recognition to disrupt oscillations, material and design development for electrodes, smaller IPGs, and improved battery design.

## 6. Conclusions

This article discusses the importance of understanding the limitations and lifespan of deep brain stimulator (DBS) devices used for treating abnormal movement disorders. The extensions and leads of both DBS and pacemaker systems are insulated to prevent the electrical current from spreading to surrounding unintended tissue including the heart and brain. However, in some cases, the insulation may wear down or develop defects over time, leading to inadequate electrical isolation. This can cause unwanted side effects or ineffective symptom management. The lead itself can experience mechanical issues, such as fracture or breakage. This can be caused by factors like repeated movements, trauma, or natural wear and tear over time. If the lead breaks, it can interrupt the electrical stimulation and lead to a loss of therapeutic effect.

To predict the lifespan of the insulated extension material, a multiple regression model was developed. This model takes into account various mechanical stress parameters including load to failure, percentage elongation, ultimate tensile strength, percentage elongation at 5 N, and modulus of elasticity. By analyzing these parameters, the model can provide an estimate of the insulated extension material’s lifespan. The development of such models that can predict the lifespan and performance of DBS leads will aid in reducing unexpected changes and ensuring the longevity and effectiveness of the treatment.

## Figures and Tables

**Figure 1 bioengineering-10-01160-f001:**
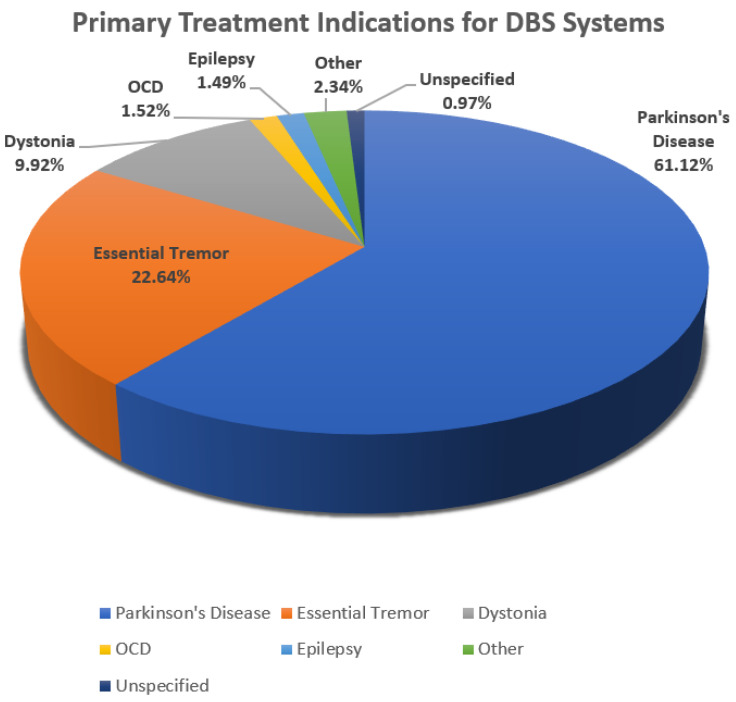
A pie chart showing the percentage of those affected by Parkinson’s disease, epilepsy, OCD, dystonia, essential tremor, and others that utilize DBS for treatment of movement symptoms [5].

**Figure 2 bioengineering-10-01160-f002:**
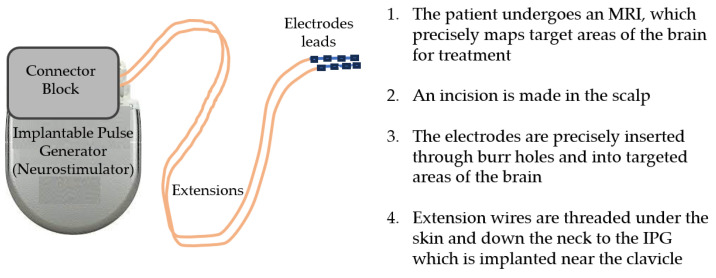
The deep brain stimulator (DBS) system comprises the implantable pulse generator (IPG), the extensions that connect the IPG to the leads, and the electrode leads that are implanted into the brain.

**Figure 3 bioengineering-10-01160-f003:**
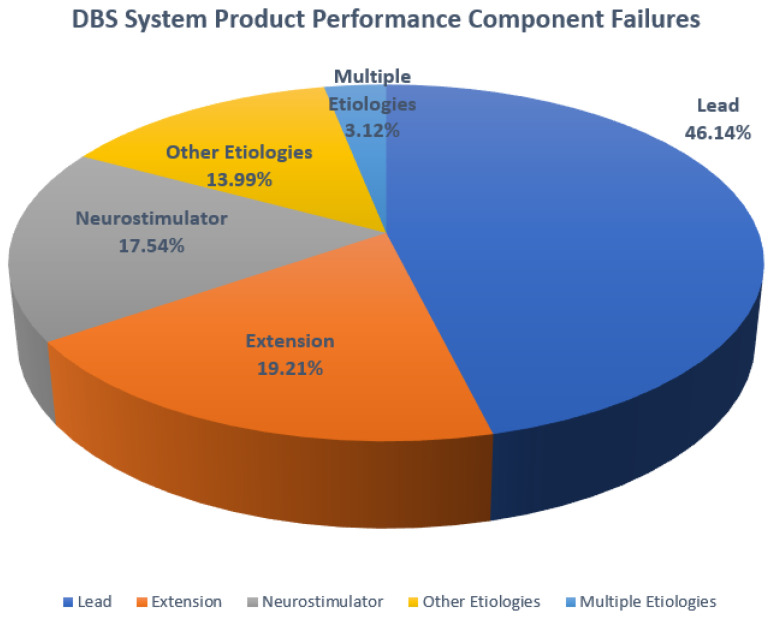
The figure shows adverse product performance events, clearly demonstrating that extensions and leads constitute greater than 50% of potential problems with a DBS system [5].

**Figure 4 bioengineering-10-01160-f004:**
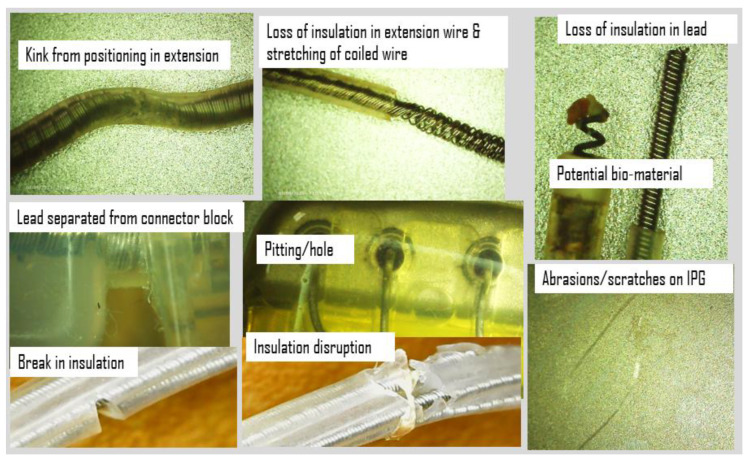
The digital microscope captured evidence of damage to the DBS system including scratches, abrasions, pitting, separation of components, eruption and loss of insulation, and evidence of biomaterials.

**Figure 5 bioengineering-10-01160-f005:**
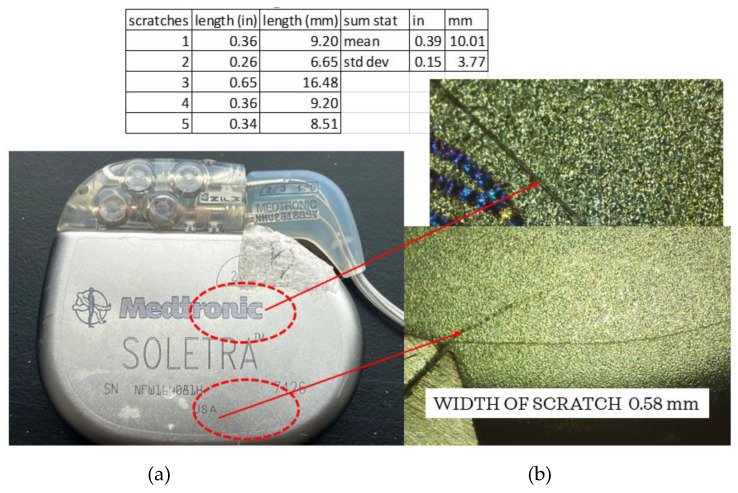
The digital microscope captured evidence of damage to the IPG case, such as scratches which had an average of ~10 mm in length and 0.58 mm in width. (**a**) is an image of the DBS device that was evaluated, and the red circles indicate the location of the scratches. (**b**) is an image of the scratches that are possibly a result of instruments used to posthumously extract the device.

**Figure 6 bioengineering-10-01160-f006:**
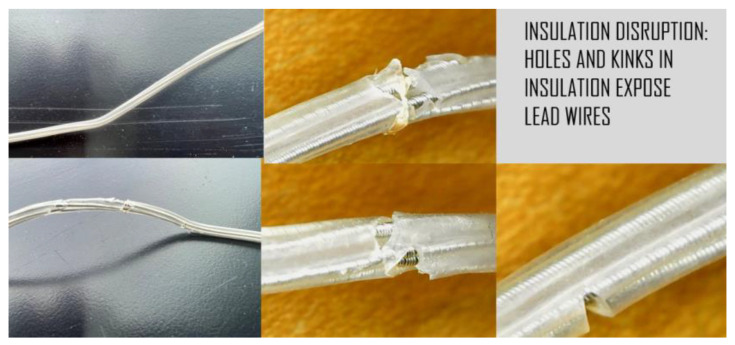
The digital microscope captured evidence of damage to the leads including corrosion, cracks, pitting, eruptions, and tears.

**Figure 7 bioengineering-10-01160-f007:**
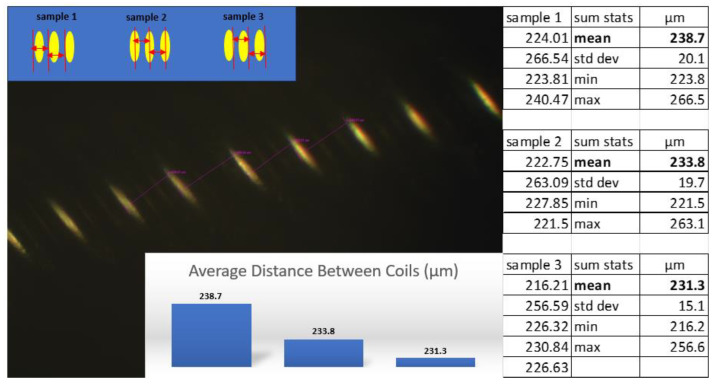
Distance measurements between coils using the inverted compound microscope. The distances between coils were made by measuring from the left edge of the coil to the neighboring left edge of the coil (sample 1); from the middle of the coil to the middle of the neighboring coil (sample 2); and from the right edge of the coil to the neighboring right edge of the coil (sample 3).

**Figure 8 bioengineering-10-01160-f008:**
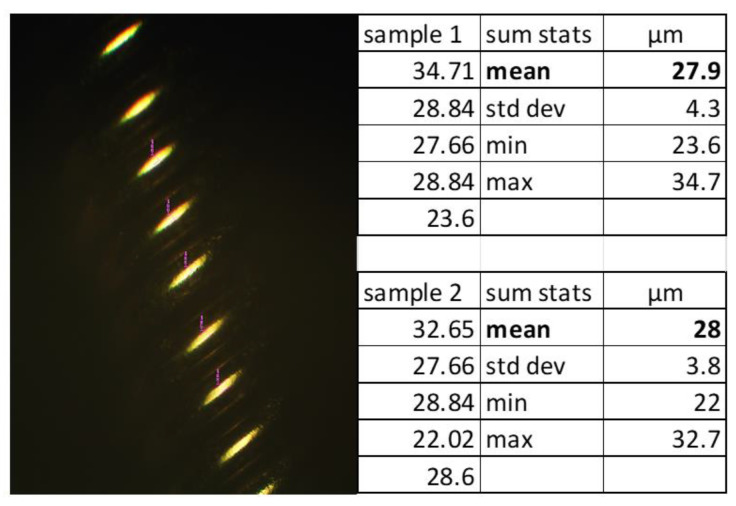
Measurement of the thickness of coils using the inverted compound microscope at two different locations.

**Figure 9 bioengineering-10-01160-f009:**
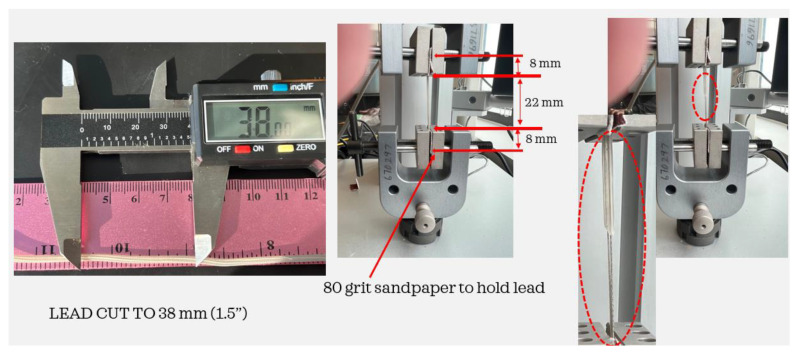
Testing tensile strength of the silicone insulation of the Medtronic extension.

**Figure 10 bioengineering-10-01160-f010:**
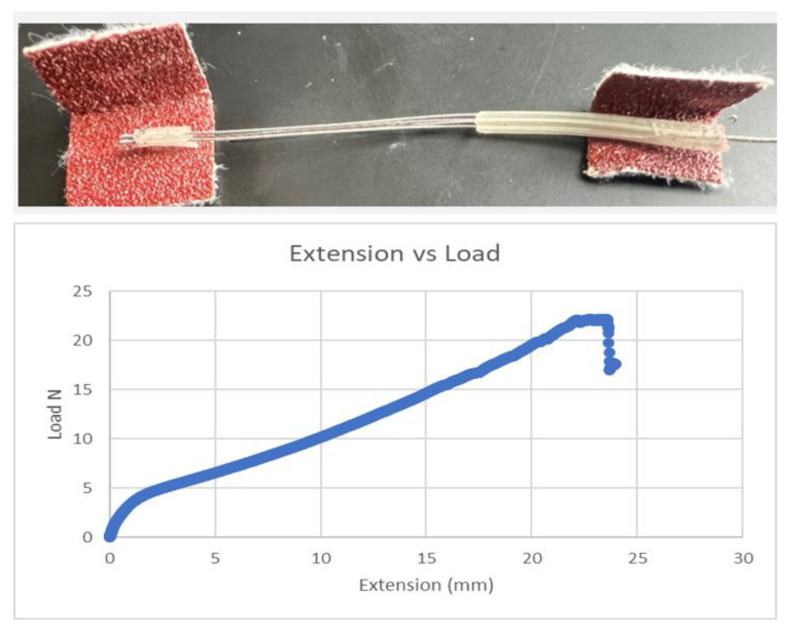
Load vs. displacement results of the extension wire protected by silicone.

**Figure 11 bioengineering-10-01160-f011:**
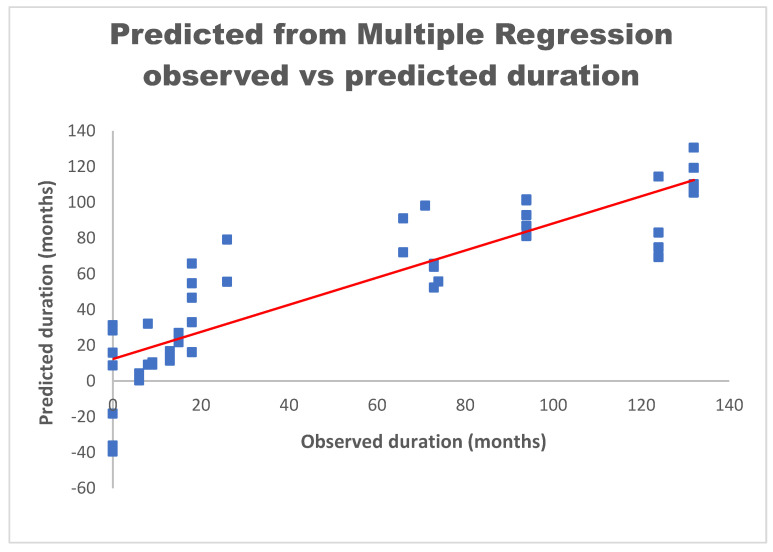
Multiple regression results of predicting duration given load to failure, maximum elongation, 5 N elongation, UTS, and modulus of elasticity.

**Figure 12 bioengineering-10-01160-f012:**
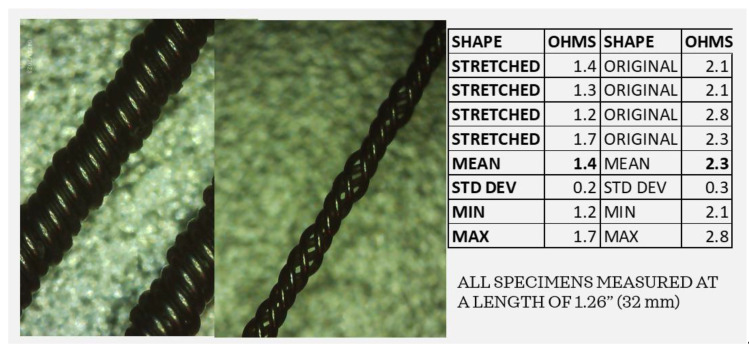
Stretching of the coils can lead to a lower resistance and, potentially, a higher current.

**Table 1 bioengineering-10-01160-t001:** The dependent variable, duration in months, is predicted using each of the measured mechanical property values [7].

Mechanical Properties	Linear Prediction (0–132 Month)
Load to Failure	R2 = 0.60
Max Elongation	R2 = 0.48
Elongation at 5 N	R2 = 0.38
Ultimate Tensile Strength	R2 = 0.38
Modulus of Elasticity	R2 = 0.22

**Table 2 bioengineering-10-01160-t002:** Factor correlation from a PCA demonstrates a strong correlation between load to failure and UTS as well as maximum elongation and elongation at 5 N and percent of variance.

	Factor Correlation Matrix					Eigenvalues and Percent of Variance				
	Factor 1	Factor 2	Factor 3	Factor 4	Factor 5	Factor 1	Factor 2	Factor 3	Factor 4	Factor 5
Load to Failure	0.51065	0.35014	−0.20999	0.08232	−0.7522	2.929	1.16954	0.4497	0.32661	0.12515
Max Elongation	0.46342	−0.22146	0.77238	0.37187	0.03659	58.58%	23.39%	8.99%	6.53%	2.50%
5 N Elongation	0.49581	−0.21709	0.03408	−0.82992	0.13453	SUM	81.97%	90.96%	97.50%	100%
Ultimate Tensile Strength	0.444	0.52863	−0.25123	0.22015	0.64174					
Modulus of Elasticity	−0.28723	0.70836	0.5432	−0.34306	−0.0544					

**Table 3 bioengineering-10-01160-t003:** Results of the Medtronic extension samples were tested using the tensile test and compared to ranges of Salih lead data.

Test	Load to Failure	Elongation to Failure	%Elongation at 5 N	Ultimate Tensile Strength	Modulus of Elasticity
Salih and Goswami (Range)	9–25	99–187%	7–25%	3–9 MPa	1.3–22 MPa
TEST 1	14.8	171%	20.90%	3.08 MPa	9 MPa
TEST 2	17	121%	15%	3.57 MPa	5 MPa
TEST 3	13.4	107%	11.80%	4.54 MPa	4 MPa

**Table 4 bioengineering-10-01160-t004:** Duration predicted via independent linear regressions of measured mechanical properties.

Prediction of Duration from Singular Tested Mechanical Properties					
Lead Samples	Load to Failure (mo)	Elongation to Failure (mo)	%Elongation at 5 N (mo)	Ultimate Tensile Strength (mo)	Modulus of Elasticity (mo)
Test 1	21.8	93.2	18.4	6.9	7.2
Test 2	21.7	108.8	18.8	6.9	7
Test 3	21.9	113.2	19.1	6.9	7
Mean	21.8	105	18.8	6.9	7.1
Std Dev	0.1	10.5	0.3	0	0.1
Min	21.7	93.2	18.4	6.9	7
Max	21.9	113.2	19.1	6.9	7.2

**Table 5 bioengineering-10-01160-t005:** Duration predicted via independent linear regressions of measured mechanical properties from the three extension specimens.

Multiple Regression Prediction	Duration (Months)
Test 1	55.2
Test 2	61.4
Test 3	89.3
Mean	68.7
Std Dev	18.2
Min	55.2
Max	89.3

## Data Availability

Not applicable.

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
