# Peer review of "Investigation of a Deep Brain Stimulator (DBS) System"

_bioengineering, 2023, doi:10.3390/bioengineering10101160_

Round 1
Reviewer 1 Report
This manuscript investigates more about Deep Brain Stimulator (DBS) System and lacks several basic informations.
As a research paper, I strongly recommend to use the following structure for paper: Introduction, Related works, Proposed section, Result and discussion, Conclusion and References.
This paper looks like a survey article at the first instance, most of the portions are filled with theory. Comparative analysis is required with other state of the art algorithms.
What is the finding of the research (Result finding) should be mentioned in the abstract section.
What is the contribution of the research, mention it in the contribution section. Mention point by point.
Include organization of the paper at the end of the introduction section.
Keywords are missing.
Figure 1 and 2: The citation is required for these images.
Figure 4,5 and 6: The cause and investigation details are not discussed properly, i recommend you to discuss the same.
Figure 12: This should be in table format.
Section 4: The result section and conclusion should be discussed separately. Don't include reference and citation in the conclusion section.
Mention the limitation of current research and future scope in the conclusion section.
Add recent year papers (2021,2022 and 2023).
Author Response
This manuscript investigates more about Deep Brain Stimulator (DBS) System and lacks several basic informations. As a research paper, I strongly recommend to use the following structure for paper: Introduction, Related works, Proposed section, Result and discussion, Conclusion and References.
We changed the format of the paper as recommended.
This paper looks like a survey article at the first instance, most of the portions are filled with theory. Comparative analysis is required with other state of the art algorithms.
The paper has been revised to include the analytic portion in the abstract and introduction. It has been compared to previous articles as well.
What is the finding of the research (Result finding) should be mentioned in the abstract section.
The findings were included in the abstract to reflect the multiple regression predictions and changes in impedance based on distorted leads.
What is the contribution of the research, mention it in the contribution section. Mention point by point.
This is included in the abstract and conclusions.
Include organization of the paper at the end of the introduction section.
Keywords are missing.
Figure 1 and 2: The citation is required for these images.
We added the citations for these images.
Figure 4,5, 6: cause and investigation details are not discussed properly, recommend discuss the same.
Discussions are included in the text.
Figure 12: This should be in table format.
We changed this to Table format.
Section 4: The result section and conclusion should be discussed separately. Don't include reference and citation in the conclusion section.
The result section and conclusions have been changed to reflect your suggestions.
Mention the limitation of current research and future scope in the conclusion section.
The limitation of this research and future plans have been included.
Add recent year papers (2021,2022 and 2023).
Additional references have been added.
Thank you for your review.

Reviewer 2 Report
The authors present an interesting article on implanted deep brain stimulation systems. Although some aspects, such as the extent to which damage was pre-existing or occurred during explanation, could not always be clarified in the end, und das this article is relevant for technicians and clinicians both in terms of the detailed analysis of the implants and the damage or abrasion. In the future, it would be interesting to find out what observations can be found in other and further deep brain stimulation implants, for example after changes of neurostimulators. I congratulate the authors on their article and recommend its publication.
Author Response
Thank you for your review.

Reviewer 3 Report
This is a not a scientific/research paper. It is a test report. The data was from one device and it was statistically insignificant to make any conclusion.
section 2.3 seemed to be the content from another paper. It should not be repeated here. The figures 10, 11 are drafts and should not be used in the submitted paper. What was "predicted duration"?
Why are the results included in the conclusion?
The paper needs proofreading. Some sentences are hard to read.
Author Response
This is a not a scientific/research paper. It is a test report. The data was from one device and it was statistically insignificant to make any conclusion.
The test included samples from a lead extracted posthumously for which the duration (months) is unknown but predicted from a multiple regression equation devised by the authors. Additionally impedance changes were measured from deformed coils as a result of mechanical testing.
section 2.3 seemed to be the content from another paper. It should not be repeated here. The figures 10, 11 are drafts and should not be used in the submitted paper. What was "predicted duration"?
This was changed to include regression from multiple parameters (n=5) for dependent (duration).
Why are the results included in the conclusion?
Results have now been included in the conclusion.
Comments on the Quality of English Language
The paper needs proofreading. Some sentences are hard to read.
Portions of the paper have been rewritten.
Thank you for your review.

Round 2
Reviewer 1 Report
The comments are well addressed.
Author Response
I worked on improving the Methods and Results sections. I appreciate your comments and I particularly found your abstract to be helpful.
Reviewer 3 Report
The revision did not make significant improvement. This paper studied one sample. It is not sufficient to draw any conclusions. It can only validate other published results. The authors should include sufficient samples to make the results significant.
The organization of the paper is not the format of a scientific paper. Section 2 should be "methods". However, it was labeled as "proposed section". It seemed that the authors did not pay due diligence in proofreading. The data in this section should be separated as the third section "results". If the conclusion section is needed, it should be concise with a few sentences. The "discussion" and "conclusion" section can be combined.
Tables 2 & 4 should be professional, not a direct copy from a spreadsheet. Figure 10 has the same issue.
The data in Figure 10 were from a separate paper and there is no need to repeat. The conclusion with citation is sufficient.
No comment
Author Response
The revision did not make significant improvement. This paper studied one sample. It is not sufficient to draw any conclusions. It can only validate other published results. The authors should include sufficient samples to make the results significant.
I made significant improvements on the write-up for the purpose of this paper which was to develop a multiple regression model and then test it on a sample. This improved the ability to predict duration in months by a difference of 70 months
The organization of the paper is not the format of a scientific paper. Section 2 should be "methods". However, it was labeled as "proposed section". It seemed that the authors did not pay due diligence in proofreading. The data in this section should be separated as the third section "results". If the conclusion section is needed, it should be concise with a few sentences. The "discussion" and "conclusion" section can be combined.
Thank you for helping me to clear this up. Hopefully, the organization and content are both improved.
Tables 2 & 4 should be professional, not a direct copy from a spreadsheet. Figure 10 has the same issue.
I fixed the Tables and Figure 10.
The data in Figure 10 were from a separate paper and there is no need to repeat. The conclusion with citation is sufficient.
I changed this as well.